Three-dimensional reconstruction and the phylogeny of extinct chelicerate orders

Garwood Russell J. 1 russell.garwood@manchester.ac.uk
Dunlop Jason 2
1 School of Earth, Atmospheric and Environmental Sciences and The Manchester X-ray Imaging Facility, School of Materials, The University of Manchester , Manchester , UK
2 Museum für Naturkunde, Leibniz Institute for Research on Evolution and Biodiversity at the Humboldt University Berlin , Berlin , Germany
De Baets Kenneth
Electronic publication date: 2014 Nov 13
Publication date: 2014
Volume: 2
Electronic Location ID: e641
Received 2014 Aug 8; Accepted 2014 Oct 8
Copyright: © 2014 Garwood and Dunlop
Copyright year: 2014
Copyright holder: Garwood and Dunlop
License: This is an open access article distributed under the terms of the Creative Commons Attribution License, which permits unrestricted use, distribution, reproduction and adaptation in any medium and for any purpose provided that it is properly attributed. For attribution, the original author(s), title, publication source (PeerJ) and either DOI or URL of the article must be cited.
License URL: https://creativecommons.org/licenses/by/4.0/

Keywords: Arachnida, Tomography, Chelicerata, Phylogeny, Palaeozoic, Fossil, Phalangiotarbida, Haptopoda

Funding: EPSRC EP/F007906/1 EP/F001452/1 EP/I02249X/1 This work was funded in part by the EPSRC (grants EP/F007906/1, EP/F001452/1 and EP/I02249X/1). This paper was completed, in part, during a SYNTHESYS visit to Museum fur Naturkunde awarded to RG. The funders had no role in study design, data collection and analysis, decision to publish, or preparation of the manuscript.

==============================
Arachnids are an important group of arthropods. They are: diverse and abundant; a major constituent of many terrestrial ecosystems; and possess a deep and extensive fossil record. In recent years a number of exceptionally preserved arachnid fossils have been investigated using tomography and associated techniques, providing valuable insights into their morphology. Here we use X-ray microtomography to reconstruct members of two extinct arachnid orders. In the Haptopoda, we demonstrate the presence of ‘clasp-knife’ chelicerae, and our novel redescription of a member of the Phalangiotarbida highlights leg details, but fails to resolve chelicerae in the group due to their small size. As a result of these reconstructions, tomographic studies of three-dimensionally preserved fossils now exist for three of the four extinct orders, and for fossil representatives of several extant ones. Such studies constitute a valuable source of high fidelity data for constructing phylogenies. To illustrate this, here we present a cladistic analysis of the chelicerates to accompany these reconstructions. This is based on a previously published matrix, expanded to include fossil taxa and relevant characters, and allows us to: cladistically place the extinct arachnid orders; explicitly test some earlier hypotheses from the literature; and demonstrate that the addition of fossils to phylogenetic analyses can have broad implications. Phylogenies based on chelicerate morphology—in contrast to molecular studies—have achieved elements of consensus in recent years. Our work suggests that these results are not robust to the addition of novel characters or fossil taxa. Hypotheses surrounding chelicerate phylogeny remain in a state of flux.

Introduction

Arachnids and their relatives (Chelicerata) form a major branch of the arthropods, with around 112,000 living species (Zhang, 2011). They also have an extensive palaeontological record, including more than 2,200 fossil species at the time of writing (Dunlop, Penney & Jekel, 2014). Chelicerates can be found back into the Cambrian (Waloszek & Dunlop, 2002; Dunlop, Anderson & Braddy, 2004), although their record through deep time is patchy and tends to be concentrated into windows of exceptional preservation such as the late Carboniferous Coal Measures and various Cretaceous and Cenozoic ambers. Currently, sixteen arachnid orders can be recognised. Twelve have living representatives: scorpions (Scorpiones), harvestmen (Opiliones), pseudoscorpions (Pseudoscorpiones), camel spiders (Solifugae), palpigrades (Palipgradi), mites (Acariformes and Parasitiformes), ricinuleids (Ricinulei), spiders (Araneae), whip spiders (Amblypygi), whip scorpions (Thelyphonida) and schizomids (Schizomida). Four arachnid orders are extinct: trigonotarbids (Trigonotarbida), phalangiotarbids (Phalangiotarbida), haptopodids (Haptopoda) and the spider-like uraraneids (Uraraneida). To this can be added two marine groups with living representatives, the sea spiders (Pycnogonida) and horseshoe crabs (Xiphosura), as well as two extinct groups which were likely to have been primarily aquatic, the sea scorpions (Eurypterida) and the rare chasmataspidids (Chasmataspidida).

Resolving relationships between the arachnid and/or chelicerate lineages remains a challenge. Important cladistic studies include the comprehensive morphological analyses of Weygoldt & Paulus (1979), Shultz (1990) and Shultz (2007), as well as numerous applications of molecular data—sometimes with morphology combined (e.g., Wheeler & Hayashi, 1998; Giribet et al., 2002; Pepato, da Rocha & Dunlop, 2010; Rehm et al., 2011). Few of these include fossil terminals (but see Giribet et al., 2002; Shultz, 2007)—despite the fact that extinct species provide a valuable source of data (Edgecombe, 2010). Several arthropod-wide analyses—both molecular and morphological—also include chelicerates (Regier et al., 2010; Legg, Sutton & Edgecombe, 2013; Rota-Stabelli, Daley & Pisani, 2013). Yet, as noted in a recent review (Dunlop, Borner & Burmester, 2014), there is still no single accepted phylogeny for arachnids and their relatives, and there are evident discrepancies between trees derived from morphological and molecular data. Dunlop, Borner & Burmester (2014) thus recognised a minimum consensus tree, i.e., supported by various methodologies, of the form (Pycnogonida (Xiphosura (Scorpiones (Araneae (Amblypygi (Thelyphonida + Schizomida)))))). This rather extensively pruned phylogeny still excludes diverse and important groups like mites, harvestmen and pseudoscorpions, and does not place any of the fossil taxa. Xiphosura was recently interpreted as paraphyletic (Lamsdell, 2013), at least with respect to Palaeozoic ‘synziphosurines’ which may include lineages eventually evolving into both crown-group horseshoe crabs and, separately, into arachnids.

Fossils have sometimes proved controversial in phylogenetic reconstruction, and for arachnids some authors simply excluded them completely (e.g., Wheeler & Hayashi, 1998). Extinct taxa offer direct evidence of early—and possibly quite different—body plans, but often have large amounts of missing data when compared to living taxa. Furthermore, scoring morphological character states in fossils involves a degree of interpretation, and objective inferences have to be made. Despite the challenges inherent in using fossils in such analyses, recent studies have demonstrated the utility and importance of doing so in a range of different analyses (Legg, Sutton & Edgecombe, 2013; Sharma & Giribet, 2014). Furthermore, in recent years the level of interpretation required has been reduced through a number of improvements in our understanding of fossil arachnid (and arthropod) data. For example, the application of various techniques such as X-ray computed tomography—especially microtomography (µCT, e.g., Garwood, Dunlop & Sutton, 2009)—has allowed the anatomy of some fossils to be reconstructed in unparallelled three-dimensional detail. For a review of such methodologies see Sutton, Rahman & Garwood (2014).

The principal aim of this study is to draw together recently published examples of well-preserved and (where possible) three-dimensionally reconstructed fossils in a phylogenetic analysis, and to augment these with novel data for two extinct arachnid orders: Phalangiotarbida and Haptopoda. Our intention is not to present a fully resolved phylogenetic tree, and we do not consider the topology recovered the sole solution to arachnid phylogeny. Rather, we use it to identify common trends and explore the impact of fossil data on tree topologies when scored into an modified version of a previously published dataset. In addition to twenty-seven newly added fossil taxa, the matrix—which is amended from that of Pepato, da Rocha & Dunlop (2010)—has sixteen new characters to capture the fullest possible range of fossil morphology. This exercise allows us to assess how robust the placement of extinct taxa is, how these impact on the relationships recovered between extant groups, and to explicitly test some earlier hypotheses from the literature. We also hope that our matrix will constitute a starting point for further studies, and provide a useful contribution into which new fossil discoveries can be integrated. Following materials and methods information, we present first the results of our tomographic reconstructions, and then results and discussion for our cladistic analysis. We subsequently discuss the impact of fossils. Character descriptions are included as Supplemental Information.

Materials and Methods

Material and tomography

All tomographic reconstructions presented in the current study are based on material from the Coseley Lagerstätte, near Dudley, Staffordshire, UK. They are thus Late Carboniferous, from the similis–pulchra zone of the British Middle Coal Measures; Duckmantian in age (ca. 315 Ma; Pointon et al., 2012), or Westphalian B using more traditional terminology. Their preservation is as three-dimensional voids—some partially infilled with kaolinite—within siderite nodules. Scans were conducted at the Natural History Museum, London on a Nikon HMX-ST 225 scanner with a tungsten reflection target.

Two specimens of Plesiosiro madeleyi (NHM I. 15899, NHM I. 7923) from the (monotypic) extinct order Haptopoda were scanned. They were selected as the most three-dimensional representatives of all the NHM specimens of this species, and NHM I. 7923 was chosen for subsequent processing as the most complete example. This was scanned at 180 kV/175 µA, with a 0.25 mm copper filter, and 3,142 projections of exposure 354 ms, to provide a reconstructed dataset with a 19.5 µm voxel size.

Material from two species of another extinct order, Phalangiotarbida, were scanned. One was not well-preserved enough to justify further reconstruction: Goniotarbus tuberculatas (BU 696, Lapworth Museum Birmingham, also Coseley). However, the NHM specimen In 22,838, the holotype of Goniotarbus angulatus, was better resolved, and revealed important limb morphology. It was last described by Petrunkevitch (1953) whose work has, in the past, necessitated significant revision (e.g., Dunlop, 1996a; Garwood & Dunlop, 2011). Accordingly this phalangiotarbid specimen was selected for further processing. The scan was conducted at 225 kV/190 µA, and without added filtration. 3,142 projections of exposure 180 ms were collected, and a reconstructed dataset with a 16.0 µm voxel size created.

Digital visualisation

Both scans were used to create three-dimensional, virtual fossils using the custom SPIERS software suite (Sutton et al., 2012) following the methods of Garwood et al. (2012). The distal limbs of Plesiosiro madeleyi were not recovered by the scan where they were truncated by the edge of the nodule. Several of the walking legs of Goniotarbus angulatus were absent. Both models were scaled, and then exported to be presented here as VAXML models (File S1). SPIERS-generated isosurfaces were then ray-traced in Blender for figures and videos (Garwood & Dunlop, 2014)—for Goniotarbus angulatus enough of the limbs were preserved to allow missing elements to be manually modelled from those present. This was achieved in Blender, and the added elements are rendered semi-transparent for clarity (Fig. 1).

Figure 1 Digital visualisations of the haptopod Plesiosiro madeleyi (NHM I7923; (A)–(D)), and phalangiotarbid Goniotarbus angulatus (NHM In 22838; (E)–(I)).

(A) Dorsal view of P. madeleyi, showing opisthosomal segmentation and prosomal shield architecture. (B) Lateral view of the anterior ventral prosoma, nearest limbs and lateral prosoma removed, showing the nature of haptopod chelicerae. (C) Ventral view, showing ventral segmentation, and divided sternum. (D) Haptopod walking leg. (E) First left walking leg of G. angulatus, showing typical segmentation. (F) Lateral view of the anterior ventral prosoma, showing the small pedipalps, median ridge, and possible chelicerae—below the resolution of the scan. (G) Fourth right walking leg. (H) Dorsal view showing median eyes and dorsal opisthosomal segmentation. (I) Ventral view showing opisthosomal segmentation and coxo-sternal region. Abbreviations: 1–10, opisthosomal segment number; as, anterior sclerite; ch, chelicerae; cx, coxa; fa, fang; fe, femur; L1–L4, walking legs 1–4; me, media eyes; mt, metatarsus; pa, paturon; pp, pedipalps; ps, pofsterior sclerite; pt, patella; ta, tarsus; ti, tibia; tr, trochanter. Scale bars: (A, C, F–I) = 3 mm; (B, D, E) = 1 mm.

Microscopy

Hand specimen photographs of Plesiosiro madeleyi are available in the redescription of Dunlop (1999). No comparable modern photographs of Goniotarbus angulatus exist. Accordingly a plate of hand specimen photographs is published herein showing the holotype, and only known specimen (NHM In 22838: Fig. 2). This was studied and photographed using a Leica MZ16A stereomicroscope and incident light. Photographs taken at multiple focal depths were combined using the software CombineZM (see Bercovici, Hadley & Villanueva-Amadoz, 2009). Photographs of the whole fossil—which was too large for the field of view—were created by manually stitching sections using the open source raster graphics editor GIMP 2.8, and figures were assembled in Inkscape 0.48. For comparative purposes specimens of a related species (Petrunkevitch, 1949), Goniotarbus tuberculatus (NHM In 31249, NHM In 18340, and NHM In 22840), were also studied.

Figure 2 Holotype and only known specimen of phalangiotarbid Goniotarbus angulatus (NHM In 22838).

(A) Dorsal view, showing prosoma and opisthosoma, and legs 4L and 2L. Proximal portions of Leg 1L are visible at the anterior of the fossil, as are the trochanters of several of the legs on the right. (B) Ventral view showing coxo-sternal arrangement and ventral opisthosomal segmentation. Proximal portions of Leg 1L, then 2L 3L and 4L are visible. (C) A close up of the sternum, anterior to the left showing five constituent plates. (D) Detail of the anterior opisthosomal segmentation, including the posterior median bulge of the prosomal shield, and associated accommodation in the anterior opisthosomal segments. (E) The posteriormost segments (7–10) fused to create a single dorsal plate, with a terminal anal operculum. Scale bars: (A, B) = 2 mm; (C–E) = 1 mm.

Character coding

The current analysis of 86 taxa and 192 characters is a modified version of ‘Matrix A’ created by Pepato, da Rocha & Dunlop (2010). A particular focus of the previous study was to clarify the position of, and relationships within, the mites. Consequently the analysis had a large number of mite-specific characters. The current study has different goals, and for ease of analysis and clarity we exclude numerous characters which are only helpful for resolving ingroup relationships within one or both of the two major mite lineages (i.e., acariforms and parasitiforms). We remove two further characters based upon the reviews of the current manuscript, which are available with this paper. In addition to these changes, we added 16 characters relevant for fossil taxa, and modified others to make them applicable to newly introduced fossil terminals. Examples of novel characters include: a prosomal shield with a meso- and metapeltidium demarcated; the presence of genal spines; the prosoma and opisthosoma forming a single functional tagma; ‘elbowed’ chelicerae in those taxa with three-segmented chelicerae; a sixth limb modified as a paddle (or pusher); a shortened first opisthosomal tergite; six abbreviated opisthosomal tergites; the absence of a sternite for opisthosomal segment one; fusion of opisthosomal tergites 7–10; a median abdominal (genital) appendage; ventral sacs; a dorsal anal operculum; eurypterid gill tracts; and development with a nymphal stage. Full character descriptions for the updated matrix are included in Supplemental Information. We have also modified a character to explicitly code ingesting solid material, rather than extra-oral digestion. The former is easier to code for fossils, often being apparent from the morphology of mouthparts, whereas the latter is more closely based on behaviour, something which is not typically preserved in fossils. Similarly the presence of opisthosomal venom glands has been altered to code for a telson with an aculeus and vesicle (the ‘sting’), the latter being verifiable in fossil scorpions. Finally, the character recording the number of cheliceral articles now has more than three articles as an option to reflect the state observed in outgroups (Haug, Briggs & Haug, 2012; Haug et al., 2012; Briggs & Collins, 1999), and some fossil horseshoe crabs (Sutton et al., 2002; Briggs et al., 2012).

Taxon selection

As noted above, the focus on mites in Pepato, da Rocha & Dunlop’s (2010) Matrix A differed from the present study. Accordingly there were a large number of acarid terminals, which we pruned for this study. We concurrently added 27 fossil taxa to the matrix. These had on average 59% missing data in comparison to 4% for extant taxa—however we do not consider this problematic on the basis of multiple publications in recent decades demonstrating that this need not result in lack of resolution, and that excluding taxa on the basis of missing data is inadvisable (Kearney & Clark, 2003; Cobbett, Wilkinson & Wills, 2007; Wiens & Morrill, 2011; Wiens & Tiu, 2012). On the basis of the reviews of the current manuscript, the artiopodan Emeraldella brocki, from the description by Stein & Selden (2012) was included as an outgroup. This has been recovered in the mandibulate stem lineage (Stein & Selden, 2012; Ortega-Hernández, Legg & Braddy, 2013), or alternatively as more closely related to the chelicerates (Legg, Sutton & Edgecombe, 2013). Two Cambrian arthropods belonging to an assemblage variously referred to as the Megacheira, or the great appendage arthropods, were added to reflect increasing evidence that these fossils may be closely related to chelicerates (e.g., Dunlop, 2006; Edgecombe, García-Bellido & Paterson, 2011; Haug, Briggs & Haug, 2012; Haug et al., 2012; but see also Legg, 2013). The megacheiran genus Alalcomenaeus was coded on the basis of a comprehensive description of A. cambricus by Briggs & Collins (1999), and recently reported neural anatomy reported by Tanaka et al. (2013) for Alalcomenaeus sp., which minimised the degree of missing data. To assess megacheiran monophyly we added a further fossil, Leanchoilia superlata, which was recently redescribed in detail by Haug, Briggs & Haug (2012).

For analyses of extant taxa only a sea spider (Pycnogonida) was selected as the outgroup as justified in Pepato, da Rocha & Dunlop (2010). To the previously coded pycnogonids we added two well-resolved Palaeozoic fossil examples—the Silurian species Haliestes dasos described by Siveter et al. (2004) and the Devonian Palaeoisopus problematicus redescribed by Bergström, Stürmer & Winter (1980). The recently discovered species Pentapantopus vogteli (Kühl, Poschmann & Rust, 2013) resembles H. dasos and also some modern pycnogonids—but as highlighted in the original publication—incomplete preservation and a limited understanding of the species’ ontogeny precluded its placement. Accordingly we have opted to omit the species from this analysis, as with the fossil of Rudkin et al. (2013), which we consider to be controversial as it is incomplete and does not reveal a number of important sea spider features. As previously noted, Lamsdell (2013) recently challenged the monophyly of the horseshoe crabs (Xiphosura) which have traditionally been interpreted as having a stem lineage (the synziphosurines) leading up to a crown-group Xiphosurida. To test this suggestion we included three of the best preserved putative synziphosurine taxa. From the Silurian we scored Offacolus kingi based on the description of Sutton et al. (2002) and Dibasterium durgae based on Briggs et al. (2012), as well as the Devonian fossil Weinbergina opitzi redescribed by Stürmer & Bergström (1981) and Moore, Briggs & Bartels (2005).

Three Silurian representatives of the extinct Eurypterida were scored: Parastylonurus ornatus based on Waterston (1979), Mixopterus kiaeri based on Størmer (1934) and Eurypterus (formerly Baltoeurypterus) tetragonophthalmus based on Selden (1981). This allowed us to assess the issue of whether sea scorpions are closely related to scorpions; which impacts on the monophyly of arachnids and the likely number of independent terrestrialisation events (Garwood & Edgecombe, 2011; Dunlop, Scholtz & Selden, 2013). From the extinct Chasmataspidida we included the Ordovician fossil Chamataspis laurencii following Dunlop, Anderson & Braddy (2004) and the Devonian Octoberaspis ushakovi after Dunlop (2002). Eurypterids have been recovered as paraphyletic with respect to chasmataspids in some studies (Shultz, 2007), as posited by Tetlie & Braddy (2004). We also wanted to test the impact of fossils on Shultz’s (2000) Stomothecata hypothesis (i.e., Scorpiones + Opiliones) and to this end we coded five Palaeozoic scorpions: the Silurian Proscorpius osborni based on Dunlop, Tetlie & Prendini (2008); the Devonian Palaeoscorpius devonicus based on Kühl et al. (2012); the Lower Devonian genus Waeringoscorpio, based on the redescription of W. hefteri and description of W. westerwaldensis by Poschmann et al. (2008); Lower Carboniferous species Pulmonoscorpius kirktonensis, coded from Jeram (1993) with lung details from Jeram (1990); and the Carboniferous Compsoscorpius buthiformis based on Legg et al. (2012a). Adding the recently described Carboniferous stem mite harvestman Hastocularis argus from Garwood et al. (2014) and closely related Eophalangium sheari (Dunlop et al., 2003) allowed more robust assessment of the extent to which fossils impact on the proposed sister group relationship between scorpions and harvestmen.

The extinct arachnid order Trigonotarbida has been recovered as sister group to the Tetrapulmonata (i.e., spiders and their closest relatives), but relationships with the rare order Ricinulei have also been suggested in the literature (Dunlop, Kamenz & Talarico, 2009, and references therein). For trigonotarbids we scored the Devonian genus Palaeocharinus spp. from specimens assigned to Palaeocharinus rhyniensis (Hirst, 1923) and Palaeocharinus hornei (Hirst, 1923) but which we consider to be synonymous. Coding was based on Dunlop (1994a) and Garwood & Dunlop (2014). We also included two Carboniferous species we have previously reconstructed using CT scans, namely Anthracomartus hindi based on Garwood & Dunlop (2011) and Eophrynus prestvicii based on Dunlop & Garwood (2014). Another extinct (Devonian—Permian) arachnid order, the probably spider-like Uraraneida, was coded on the basis of Selden, Shear & Sutton (2008) and Selden, Shear & Bonamo (1991). Finally, the two remaining extinct arachnid orders were coded based on the digital visualisations presented herein. Characters for the Carboniferous Haptopoda derive from the model of Plesiosiro madeleyi and the previously published account of Dunlop (1999). Phalangiotarbida coding was again based on the model presented herein for Goniotarbus tuberculatus, plus data from Pollitt, Braddy & Dunlop (2004) for Bornatarbus mayasii (both Carboniferous).

Cladistic analysis

The matrix was analysed with TNT v.1.1. (Goloboff, Farris & Nixon, 2008; made available with the sponsorship of the Willi Hennig Society), using unordered multistate characters, and traditional search options. Searches comprising tree bisection-reconnection [TBR] with 1,000 replicates, saving 100 trees per cycle were conducted on the full matrix (File S2), and a pruned version of the matrix excluding fossil taxa (File S3). The data matrix is also available in the public database Morphobank (http://www.morphobank.org; Project 1274). For equally weighted analyses (EW) with fossils, TNT was used to create strict consensus trees which were exported as SVGs into Inkscape, and numerous analyses were run to explore the data with differing taxa and characters excluded to explore their impact. Results for equally weighted analyses lacking fossils were exported as .tre files of the strict consensus, and trees collapsed in Figtree 1.4.1 before being exported to Inkscape. Analyses were also run using implied weighting (IW) to assess the impact of homoplasy on the results. Goloboff (1993) and Goloboff et al. (2008) provide an overview of this weighting scheme, whilst Legg, Sutton & Edgecombe (2013), Legg & Caron (2014) and Ortega-Hernández, Legg & Braddy (2013) provide justification of its use in a palaeontological context. We note, however, the comments of reviewer #1 of the current manuscript—available with the paper—criticizing this weighting scheme; no peer-reviewed contribution discussing these issues is currently available in the literature. Due to a number of difficult-to-place groups (Phalangiotarbida, Ricinulei, Parasitiformes) and resulting instability, when run with a variety of concavity constants (k = 0.25, 1.0, 3.0, and 10.0) tree topology changed. Here we present a strict consensus of the most parsimonious trees for each concavity constant. For the analyses including fossils, resampling was carried out in TNT: we provide jackknife (Farris et al., 1996; 33% removal probability, 1,000 replicates), bootstrap (Felsenstein, 1985; 1,000 replicates) and Bremer support (Bremer, 1994) values for the equal weights tree. Nodal support values of the first two of these are shown as absolute frequencies. For implied weights trees we show support through symmetric resampling—chosen because it is unaffected by character weighting (Goloboff et al., 2003)—using a change probability of 33%, and 1000 replicates, and reporting absolute frequencies.

Tomography Results

Reconstruction of Plesiosiro madeleyi

The digital visualisation of haptopodid Plesiosiro madeleyi (NHM I.7923; Figs. 1A–1D) presented herein largely corroborates previous work on this species (Pocock, 1911; Petrunkevitch, 1949; Dunlop, 1999). Some elements—such as distal limb articles—are not resolved in the CT scan as they run along the crack in the nodule. The most complete leg is shown in Fig. 1D. Accordingly we refer the reader to Dunlop (1999) for these details—which include a full description and measurements of the scanned specimen—and focus here on clearly resolved and/or novel anatomical elements. Note that the scanned specimen shows a small amount of distortion due to lateral compression.

As previously reported, the posterior margin of the prosomal shield terminates with a posteriorly directed ridge, obscuring some of tergite one (Dunlop, 1999; see File S1, animation in File S4). Clipping the digital visualisation provides no clear evidence for any kind of locking structure between the prosomal shield and the first tergites, such as is seen in the extinct trigonotarbids for example. Instead the prosoma-opisthosoma junction in Plesiosiro madeleyi forms a simple ‘z’-shaped arrangement in lateral section. Median eyes are resolved as depressions either side of a dorsal median ridge on the prosomal shield (Fig. 1A), reflecting the same observation in hand specimens. This is unusual for arachnids—in which the median eyes are normally raised structures—and may be a taphonomic artefact caused by the eyes inverting prior to fossilisation (see also remarks in Dunlop, 1999). The lateral prosomal shield tubercles are shown in this specimen to be broader than the rounded structures previously described, being 0.8 mm long latero-posteriorly directed ridges, whose dorsal surface projects anteriorly at the anterior prosomal shield margin in parasagittal section. They have been interpreted as possible lateral eye tubercles, but evidence of explicit lenses is lacking. It has also been speculated that Plesiosiro madeleyi was a harvestman (see below), but these lateral tubercles also showed no obvious openings for repugnatorial glands; as would be expected if these structures were raised ozophores similar to the condition in cyphophthalmid harvestmen. Overall, the results of the phylogeny presented herein support Dunlop (1999) in the suggestion that these projections probably represent lateral eye tubercles. Immediately posterior to the tubercles are small depressions.

The ventral prosoma is well-resolved, and confirms the presence of anterior and posterior sclerites in the sternum (Fig. 1C), the former bearing an anterior pair of protrusions. Significantly, the scan unequivocally demonstrates chelicerae of a ‘clasp-knife’ type, comprising a proximal (minimum of 0.7 mm in length) and distal (0.9 mm) article (Fig. 1B). There is no evidence of a third cheliceral article as reported by Petrunkevitch (1949). The chelicerae are ventral to the median anterior projection, their attachment being aligned essentially level with the median eyes. Palpal coxae cannot be resolved due to the crack in the nodule, but the model suggests that the chelicerae were probably tucked between the bases of the pedipalps in life. The chelicerae are preserved with the proximal article dorsally oriented, with a geniculate joint, and the distal article ventrally directed. Thus they probably had something approaching an ‘orthognath’ bite (i.e., hinged so cheliceral movement is parallel to the sagittal plane), similar to the mesothele and mygalomorph spiders (Kraus & Kraus, 1993). The opisthosomal segmentation pattern for Plesiosiro madeleyi resolved here—i.e., 12 segments in total—corroborates that reported by Dunlop (1999).

Reconstruction of Goniotarbus angulatus

Digital reconstruction of the holotype of phalangiotarbid Goniotarbus angulatus (NHM In 22838) reveals an arachnid with a broad prosoma-opisthosoma boundary (Figs. 1H and 2A, File S1, animation in File S4). The total body length is 17.0 mm whereby the prosomal shield is 6.5 mm long and 7.1 mm wide at its curved posterior margin. A median bulge causes crowding of—and obscures in part—the anterior-most opisthosomal tergites (Fig. 2D). The prosomal shield has an anteriorly positioned median eye tubercle, however the eye arrangement is not readily apparent—perhaps the reason previous works reported either two (Pocock, 1911) or six (Petrunkevitch, 1953) lenses. Careful study suggests that are three depressions in a triangular arrangement on one side—a number matching the state observed in other phalangiotarbids (Pollitt, Braddy & Dunlop, 2004)—however, this isn’t seen on the opposing side of the prosomal shield, and thus we treat the observation with caution, and discuss the impact of different codings below. The dorsal surface of the prosomal shield is demarcated by three pairs of radiating linear depressions. These are not so clearly visible towards the middle, but become increasingly pronounced laterally (File S1). The median prosomal shield bears a subtly raised region with concave lateral margins, widening anteriorly towards the eyes and also posteriorly.

The opisthosoma is 10.6 mm long and maximally 7.5 mm wide. Opisthosomal segments 1–6 are very short and closely spaced, as is typical for members of this extinct order (Figs. 1H and 2D). Tergites 1 and 2 express anterior curvature at their edges, accommodating the curved prosomal margin (see above). Tergites 4–6 have straighter margins, and 6 is slightly longer than the five preceding tergites (Fig. 2D). Other data (e.g., Pollitt, Braddy & Dunlop, 2004) suggest that the phalagiotarbid ground pattern was an opisthosoma with 10 clearly expressed tergites dorsally. In Goniotarbus angulatus (and many other species) tergites 7–10 appear to be fused into a single dorsal plate covering the back end of the opisthosoma. However, in NHM In 22838 this original segmentation is marked by subtle, v-shaped linear depressions (Fig. 2E). The presence of a posterior depression to demarcate the original tergite ten corroborates the observation reported by Petrunkevitch (1953), which was missing in the original description of Pocock (1911). The posterior opisthosoma bears the dorsal anal operculum; marked by a pit in NHM In 22838 with possible discharge visible in the CT scan (Figs. 1H, 2A and 2E). This might suggest a small degree of decay prior to fossilisation.

The ventral surface is well-resolved (Figs. 1I and 2B). Scans reveal a pronounced anterior median ventral ridge near the expected position of the chelicerae (Figs. 1F and 1I), tucked between the small palpal coxae which are not visible in the hand specimen. The four triangular leg coxae are large, and all abut the sternum. The mesal surfaces of the first coxae (length 2.2 mm) are separated by the aforementioned ridge. Coxal margins otherwise appear to be in contact with the surrounding coxae (Figs. 1I and 2B). The coxae increase in size posteriorly—coxae 4 are 3.2 mm long—but these too abut sternum medially. The sternum itself (Fig. 2C) is subdivided into five plates in a ‘1–2–2’ arrangement from anterior to posterior. The ventral opisthosoma comprises the short sternites 1–4 which are crowded anteriorly between the coxae of leg 4 (Fig. 2B, total length: 1.8 mm). Sternite 5 is significantly longer than the preceding sternites, and all remaining sternites increase in length posteriorly to number 9 (length 3.9 mm). All sternites have straight posterior margins, and are divided longitudinally into three plates by two suture lines running from the distal termination of coxae four, and curving outwards to terminate at the opisthosomal margin towards the posterior end of sternite nine (Fig. 2B). Some previous studies have noted possible openings for (?tracheal) spiracles among the anteriormost sternites (Dunlop & Horrocks, 1997, Fig. 2), but these could not be identified unequivocally here. There are, however, two enigmatic raised structures on sternite 5 whose identity and function remains uncertain.

The preservation of the limbs is patchy. Based on the very small pedipalp size we presume that the phalangiotarbid chelicerae must have been tiny. This would explain their poor resolvability in the current scans. At the anterior margin of the previously described ridge between the first coxae there is a ventrally projecting feature (Fig. 1F). Further details are unrecovered due to lack of resolution, but this could represent a small pair of chelicerae. The pedipalps are also small, hanging ventrally beneath the anterior margin of the prosomal shield; although it should be noted that the exact prosomal shield margin is hard to differentiate from the crack along which the nodule was split in this specimen, and thus it is possible they projected anterior of the prosomal shield in life. Individual articles in the pedipalps could not be resolved in the CT scan. The first right walking leg is truncated midway along the tibia. All other legs on this side are truncated at the trochanter-femur joint. The legs on the left side are more complete (Figs. 1E and 1G). On the basis of the preserved articles, leg length appears to increase posteriorly from 1 to 4. Limb article proportions are generally similar throughout—on the most complete leg (left leg 1, Fig. 1E) the measurements are: trochanter, 1.0 mm; femur, 1.6 mm; patella, 1.7 mm; tibia, 1.6 mm; metatarsus, 1.1 mm and tarsus, 1.2 mm. Details of any terminal claws (apoteles) on the legs are equivocal.

Cladistics Results and Discussion

Analysis using traditional search options (TBR) and equal weights (EW) resulted in 383 trees of 455 steps (average weighted character fit, WCF, 101.5; Goloboff, 1993), presented here as a strict consensus (Fig. 3). Using implied weights (IW), results differed with changing concavity constants: k = 0.25 resulted in 3 trees of 80.38824 steps (all with WCF = 107.6); k = 1.0 in 3 trees of 60.05750 steps (all with WCF = 127.94); k = 3.0 in 12 trees of 37.26555 steps (all with WCF = 150.73); and k = 10.0 in 6 trees of 16.71747 steps (all with WCF = 171.28). A consensus of the MPTs for each k value is shown in Fig. 4. Without fossil taxa, 40 trees of 384 steps were recovered under the same EW search parameters (average WCF 150.9). In general, several relationships are consistent across all search parameters and our results—and their implications—are discussed in further detail below.

Figure 3 Results of the cladistic analysis presented herein under equal weights analysis.

The trees show the strict consensus of equally weighted analyses of the matrices presented here (File S2, File S3, morphobank project 1274). (A) Tree showing the analysis results with fossils included. Bremer, jackknife and bootstrap support values are provided for each node as shown in the key. (B) Tree recovered with fossil terminals removed—ordinal clades are collapsed for clarity.

Figure 4 Results of the cladistic analysis presented herein under implied weights.

The trees show the strict consensus of implied weights analyses of the matrices presented here (File S2, Morphobank project 1274). Symmetric resampling support values are provided on the basis that these are unaffected by character weights. (A) The topology for concavity constants (k values) 0.25 and 1.0, which are identical. k = 0.25 support value above each node in red, k = 1.0 below in grey. (B) Tree for k = 3.0. (C) Topology for k = 10.0.

Chelicerata

The two analyses including fossils are rooted on the artiopodan Emeraldella brocki. In the EW tree (Fig. 3) the megacheiran taxa form a clade, sister group to (Eurypterida + (Chasmataspida + Xiphosura)). This relationship—defined through the presence of lateral eyes and a pair of median eyes—has essentially no support in the current analysis. In all IW analyses these taxa are found either in a polytomy with all remaining taxa, or as a clade sister group to these (Fig. 4). All IW analyses thus recover Chelicerata (i.e., those taxa which were traditionally assigned to Pycnogonida, Merostomata and Arachnida; Dunlop, 2010)—a relationship with stronger support, and in keeping with the idea that Alalcomenaeus and L. superlata could be considered possible stem-chelicerates in the literature (e.g., Edgecombe, García-Bellido & Paterson, 2011; Haug, Briggs & Haug, 2012; Haug et al., 2012; although see also Legg, Sutton & Edgecombe, 2013). We highlight, however, that due to a lack of further Cambrian fossil taxa we do not consider this a robust test of megacheiran relationships. Megacheirans were not our focus group, and study of a range of early Palaeozoic arthropods is needed to resolve the origins of chelicerates and their probable stem-group in detail. We refer to Chen, Waloszek & Mass (2004), Haug, Briggs & Haug (2012), Haug et al. (2012), Legg et al. (2012b), Legg (2013) and Legg, Sutton & Edgecombe (2013) for work in this direction. At the base of the chelicerates in IW analyses we find a clade comprising Dibasterium durgae and Offacolus kingi as sistergroup to all other chelicerates. This position for O. kingi and D. durgae results from their chelicerae, which possess more than three articles, and the presence of exopods on all the postcheliceral prosomal appendages, shared with the Cambrian taxa included, but no other chelicerates. Under EW this clade is sister group to the arachnids and pycnogonids instead: another grouping with essentially no support, in which we place little confidence.

Merostomata?

Excluding Dibasterium durgae and Offacolus kingi, the remaining horseshoe crabs in our analysis resolved in a clade together with the extinct eurypterids and chasmataspidids. In EW analysis, this group is defined by the increase in head shield segments and reduction of cheliceral segments to three relative to the Megachiera and outgroup, whilst with IW these share the synapomorphy of a cephalic doublure (where known). In IW analyses this clade is sister group to an (Arachnida + Pycnogonida) clade (see below), whilst EW analyses have synziphosurine taxa in this position, as previously highlighted. Using EW Chasmataspidida, the remaining Xiphosura, and Eurypterida each form a monophyletic group, with the relationships (Xiphosura (Chasmataspidida + Eurypterida)). Under IW, the position of the synziphosurine Weinbergina opitzi is unstable—at high k values (k = 3.0, k = 10.0) it is sister group to (Eurypterida (Chasmatsapidida (modern Xiphosurida)), whereas at lower values it is resolved as sistergroup to (C. laurencii + modern Xiphosurida). At all k values, the eurypterids form a clade, whilst chasmataspids are either split by W. opitzi, or form a grade to modern Xiphosura. In an extant-taxa-only analyses under EW and IW, rooted on pycnogonids, Xiphosurida form a monophyletic sister clade to Arachnida.

This overall result with fossils included is interesting in that horseshoe crabs and eurypterids were traditionally grouped together as the class Merostomata; the aquatic counterpart to a largely terrestrial class Arachnida. As argued by authors like Kraus (1976), this is primarily an ecological distinction rather than a phylogenetic one. Previous cladistic studies consistently failed to recover Merostomata and usually placed eurypterids closer to arachnids instead (Weygoldt & Paulus, 1979; Shultz, 1990; Shultz, 2007; see also below). We also failed to recover Merostomata in its traditional sense, since our synziphosurines are paraphyletic, with D. durgae and O. kingi recovered elsewhere. In IW analyses we are, however, left with a monophyletic unit comprising remaining Xiphosura sensu stricto (albeit here including the synzophosurine W. opitzi; see below), Eurypterida and Chasmataspidida. Were this clade to prove robust—but see Lamsdell (2013) for an alternative model—the name Merostomata remains available for this taxon.

Xiphosura/Xiphosurida

Lamsdell (2013) analysed many of the Palaeozoic fossils traditionally assigned to horseshoe crabs, concluding that Xiphosura is not monophyletic and that the fossils placed here actually comprise a paraphyletic assemblage of basal chelicerates and stem taxa for the lineage leading up to modern Xiphosurida. Our taxon coverage was not nearly as extensive as Lamsdell’s, but in concordance with his study we also failed to recover a monophyletic Xiphosura. Both here under IW (but not EW, see above) and in Lamsdell (2013) Offacolus kingi resolved in a basal position among Chelicerata (Fig. 4). On the basis of our support values, we certainly consider this the stronger of the two possibilities we present herein. Lamsdell did not score Dibasterium durgae—the species was described while his paper was in press—but added a comment in proof addressing the fact that the original description by Briggs et al. (2012) recovered a monophyletic Xiphosura, a result repeated in Legg, Sutton & Edgecombe (2013). Lamsdell (2013) questioned a number of the characters in this matrix. The IW results presented herein agree in part with those of Lamsdell, in recovering a paraphyletic synziphosurine grade at the based of the chelicerate tree. Our analysis differs from Lamsdell’s scheme in recovering, at least under some parameters of analysis, the synziphosurine Weinbergina opitzi in a clade together with the living horseshoe crabs. In his 2013 paper Weinbergina was part of a newly formulated Prosomapoda assemblage; essentially comprising the sister group of the horseshoe crabs sensu stricto, the eurypterids and the arachnids. Lamsdell’s hypothesis that synziphosurines include basal chelicerates—rather than just basal horseshoe crabs—has much to recommend it, and the position(s) recovered, but we also note that whilst two cladistic analyses have now recovered this result, both are limited in their taxon sampling in comparison to, for example, Legg, Sutton & Edgecombe (2013). Only through continued analysis and work on fossil chelicerates can light be shed on this issue. A Xiphosurida crown group—represented here by the two living horseshoe crabs—was, unsurprisingly, recovered as monophyletic.

Eurypterida

The most recent evidence in favour of eurypterids being the sister group of the arachnids was published by Kamenz, Staude & Dunlop (2011), who identified the enigmatic ‘horn organ’ in the opisthosoma of an exceptionally preserved eurypterid as a potential precursor of a spermatophore. Sperm transfer via spermatophores was thus proposed as a putative synapomorphy for (Eurypterida + Arachnida) and the name Sclerophorata was introduced by Kamenz, Staude & Dunlop (2011) for this clade; nomenclature also followed by Lamsdell (2013). Despite this, Eurypterida did not resolve in the present study together with Arachnida, rather they were drawn into the previously discussed merastomatid clade through their cephalic doublure (a character missing in Dibastierum and Offacolus, but present in modern xiphosuran taxa), and ‘elbowed’ chelicerae where known. Eurypterids were recovered as monophyletic in all analyses, with the internal relationships (Parastylonurus ornatus (Eurypterus tetragonophthalmus + Mixopterus kiaeri)). Until recently there were two very good synapomorphies for Eurypterida: a median abdominal (or genital) appendage on the underside of the opisthosoma and a plate-like metastoma covering the back end of the coxal gnathobases. Both these characters were subsequently identified by Dunlop (2002) in at least one genus of chasmataspidid (including Octoberaspis), but the group is defined here by the fusion of the anteriormost abdominal appendages to form a large genital operculum.

Chasmataspidida

In the EW analysis, the chasmataspidids are recovered as monophyletic on the basis of genal spines in both coded species, and this clade is in turn sister group to the eurypterids. The chasmataspidid-eurypterid clade is defined by the combined presence of a metasoma, twenty-segmented body, and presence of a median abdominal appendage. Eurypterids and chasmataspidids also share a very short first opisthosomal tergite. In IW analyses, chasmataspidids are recovered as a grade relative to the extant Xiphosura at higher k values (k = 3.0, k = 10.0), or split by W. opitzi. Neither IW or EW topologies have strong support. That the group is paraphyletic is in accordance with Tetlie & Braddy (2004) and Shultz (2007), but contra Dunlop, Anderson & Braddy (2004). The IW results reflect a fundamental conflict relating to chasmataspidids: in addition to the similarities between some chasmataspidids (e.g., Diploaspis casteri, Octoberaspis ushakovi) and eurypterids, other taxa more closely resemble horseshoe crabs (e.g., Chasmataspis laurencii, which is found as sister group in IW analyses to Xiphosurida). Indeed when Chasmataspis laurencii groups with xiphosurids it does so on the basis of the presence of a cardiac lobe. We note, however, that all chasmataspids described to date share a similar and distinctive body plan in which the opisthosoma consists of a short preabdomen and long nine-segmented postabdomen. These synapomorphies are outweighed in the IW analysis by those shared with xiphosurans.

Pycnogonida

One result of the present analysis merits particular comment, namely the position of Pycnogonida. The sea spiders usually resolve either as sister group of all other chelicerates—which collectively form the Euchelicerata sensu Weygoldt & Paulus (1979)—or sometimes as sister group of all other arthropods in studies with a broader taxon sampling; see Dunlop & Arango (2005) for a review. In the present study, in the trees including fossil taxa, Pycnogonida are monophyletic. This is to be expected as there are many synapomorphies for both fossil and living sea spiders such as the proboscis used for feeding or the unique oviger appendage. However in the current analysis, Pycnogonida resolved in a more derived position based on our dataset, specifically as the sister group of Arachnida. This result is controversial, but has precedence among some mitochondrial DNA studies (e.g., Podsiadlowski & Braband, 2006; Jeyaprakash & Hoy, 2009) which also recovered an ingroup position for sea spiders. The work of Simon & Hadrys (2013) makes apparent the problems of using mtDNA for deep divergences in arthropod taxa, and indeed, Dunlop, Borner & Burmester (2014) cautioned that the mtDNA results in question may be an artefact. We strongly suspect that is the case here. The position of of pycnogonids in this study probably results from long branch attraction, coupled with taxon selection. The pycnogonid clade has a multitude of synapomorphies, whilst the effect of the outgroup is to place synziphosurine and merastomatid taxa at the base of the tree. The limited number of completely preserved pycnogonid taxa precludes breaking up this long branch with such fossils. Rather this issue could be addressed in future studies by adding further outgroup taxa sampling the origins of, and early splits within, the total group chelicerates. These should include a range of Cambrian arthropods. In addition to further megacheirans and xenopods, definitively mandibulate taxa would be beneficial, as could be adding a non-arthropod outgroup, or including novel characters suites to better test the polarity of pycnogonid characters. Within the pycnogonids under EW the fossil taxa are recovered as a grade to a polytomy comprising extant species, whereas in IW fossil taxa are recovered as an internal clade due to multi-segmented chelicerae—the remaining taxa are a polytomy. In extant-taxa only trees Pycnogonids are not monophyletic as we have chosen to root the tree with a single species as the outgroup.

Arachnida

The present dataset universally recovers Arachnida as a monophyletic group, sharing the apomorphies across all analyses of lost appendages (at least in postembryonic instars) on opisthosomal segment 1, and a tibial origin of the apotele depressor. In IW analysis this includes the presence of slit sense organs (albeit secondarily absent in Palpigradi). All arachnid orders with more than one taxon included were found to be monophyletic, with a single exception, highlighted below. A monophyletic Arachnida is in concordance with the majority of the published morphological (Weygoldt & Paulus, 1979; Shultz, 1990; Shultz, 2007) and molecular (Wheeler & Hayashi, 1998; Regier et al., 2010) analyses. However, as outlined below, relationships between the arachnid orders are dependent on analytical parameters, and differ between the trees we present.

Despite potential synapomorphies for a scorpion + eurypterid clade—such as a five-segmented postabdomen (e.g., Dunlop & Webster, 1999)—this was not our most parsimonious result. The similarities between scorpions and sea scorpions would thus be homoplastic. This is supported by the results of the only published eurypterid phylogeny (Tetlie, 2007; a tree based on an unpublished phylogeny from the author’s PhD thesis), who found that the most scorpion-like genera with strongly raptorial forelimbs and curved telsons (Mixopterus, Carcinosoma) resolve in a derived position within Eurypterida; again indicative of convergent character aquisition.

Opiliones, Pseudoscorpiones, Scorpiones, Phalangiotarbida

In all analyses excluding IW with low k values (k = 0.25, k = 1.0), we recover a weakly supported clade of the form ((Pseudoscorpiones + Scorpiones) + (Phalangiotarbida + Opiliones)). With extant taxa only under EW, a clade of the form (Opiliones (Scorpiones + Pseudoscorpiones)) is recovered, whilst in IW extant only analyses and with fossils and low k values, Stomothecata (i.e., Opiliones + Scorpiones) is recovered. In this case the relationship for the other members of the clade is (Phalangiotarbida + (Pseudoscorpiones + (all other non-stomothecata arachnids))). Synapomorphies for ((Pseudoscorpiones + Scorpiones) + (Phalangiotarbida + Opiliones)) include bicondylar femoropatella articulation, bicondylar patellotibial articulation, and the presence of a cheliceral tergal–deutomerite muscle. We note, however, that these characters are unknown for fossil taxa, including both phalangiotarbid species. In a wider context, the position of Scorpiones has been one of the most divisive issues in arachnid phylogeny. These animals have been variously interpreted as closely related to eurypterids (see above), as sister group to all other arachnids (Pocock, 1893; Börner, 1904; Weygoldt & Paulus, 1979) or more recently as the sister group of Opiliones (Shultz, 2000; Shultz, 2007) based on the shared presence of a preoral chamber formed from projections of the anterior leg coxae (the stomotheca, hence Stomothecata). A recent alternative proposal from Sharma et al. (2014) is Arachnopulmonata—a sister group relationship between the scorpions and tetrapulmonates (spiders and their relatives, a group introduced below). We recover Scorpiones and Opiliones within a clade, but with Pseudoscorpiones and Phalangiotarbida included. The novel clade (Pseudoscorpiones + Scorpiones) is an interesting result, because although scorpions and pseudoscorpions look superficially similar, features like their large pedipalpal claws are generally interpreted in modern analyses as homoplastic. Some (non cladistic) studies did place pseudoscorpions and scorpions together, with Savory (1971) introducing the name Scorpionides for these two orders, but again this was largely based on inferences from gross morphology rather than explicit and testable synapomorphies. Pepato, da Rocha & Dunlop (2010) recovered (Scorpiones + Pseudoscorpions) under some parameters of analysis and indeed our tree builds upon their morphological dataset. In our EW analysis they are united by the nature of their palpal chelae, presence of a patellotibial extensor, ventrally/posteroventrally orientated anterior transpatellar muscle insertion, loss of the posterior patellotibial muscle, and isolecithal/telolecithal eggs.

In most other cladistic studies (Weygoldt & Paulus, 1979; Shultz, 1990; Shultz, 2007; Wheeler & Hayashi, 1998; Giribet et al., 2002) Pseudoscorpiones were placed as the sister group of Solifugae, together forming the clade Haplocnemata; a name introduced by Börner (1904). Putative synapomorphies identified for Haplocnemata include a very short femur and a correspondingly long patella—in some schemes they were named Apatellata because of confusion about whether these arachnids even had a proper ‘knee’ joint—and two pairs of tracheae opening on the third and fourth opisthosomal segments. We do not recover Haplocnemata in any of our analyses. A number of morphological characters found in common between Pseudoscorpions and Parasitiformes among the mites do not result in the grouping of these taxa in any analyses. In our low k values (k = 0.25, k = 1) analyses we recover stomothecata in a form whereby many of the fossil scorpion taxa form a polytomy at the base of a clade comprising the remaining scorpions and the harvestmen. This clade is a polytomy between the fossil scorpion Compsoscorpius buthiformis, the crown group Scorpiones, and the Opiliones. A plesiomorphic scorpion grade leading to crown Stomothecata is novel, but only appears under a limited range of analytical parameters.

Apart from the low k values analyses where they are sister group to the non-Stomothecata arachnids, the Phalangiotarbida are recovered as sister group to the Opiliones. This grouping has low support, and is supported only by the number of body segments in the EW analysis. Previous hypotheses of relationships with Opiliones were put forward by Petrunkevitch (1948)—and this is the first cladistic support for this hypothesis. Alternatives of close relationships with the Opilioacariformes among the mites (Dunlop, 1995) or with the tetrapulmonate arachnids (see below; Pollitt, Braddy & Dunlop, 2004) were not supported here. We are cautious of this result however, which we believe lacks stability—in part, we suspect, because of a lack of synapomorphies phalangiotarbids share with other orders. Phalangiotarbids are unusual-looking creatures and preserve a series of novel autapomorphies—short tergites, divided sternites, dorsal anal operculum, etc.—none of which suggest an animal close to the morphological ground pattern of the arachnids. Furthermore, important questions regarding the group’s anatomy remain unclear: for example the nature of their eyes. Many phalangiotarbid taxa possess six eyes located on a median ocular tubercle. Pollitt, Braddy & Dunlop (2004) tentatively identified these as three pairs of lateral eyes. However, phalangiotarbid eyes could equally represent one median eye pair, and two lateral pairs located in close proximity to each other. We have coded the taxa as having the latter—many spiders have a single tubercle bearing two median plus lateral eyes—but note the exact nature of the eyes is impossible to ascertain in extinct groups. If phalangiotarbids are coded as possessing three pairs of lateral eyes, they are recovered under EW in a basal polytomy of non-palpigrade arachnids, along with clades comprising Opiliones, (Pseudoscorpiones + Scorpiones), Pantetrapulmonates, and a clade comprising all remaining taxa (Ricinulei + (Parasitiformes + (Solifugae + Acariformes))). With IW under this coding they match the position seen in the low k values trees presented herein. Unfortunately even tomography could not resolve key features such as the nature of the chelicerae (chelate or ‘clasp-knife’?) that could support a more robust placement. In order to achieve any certainty, widespread application of these techniques to currently known phalangiotarbids, or new fossil discoveries, will be key.

Palpigradi

Palpigradi are enigmatic arachnids which appear to retain a suite of plesiomorphic character states, such as chelicerae with three articles, multiple claws on the pedipalp and a telson. Although superficially resembling whip scorpions (Thelyphonida) they are widely perceived as ‘primitive’ arachnids and tend to emerge as an early branching clade. A definitive position is hard to resolve and previous studies showed little consistency in their results. We add no further certainty here: the group resolves variously as sister group to all other arachnids (EW)—reflecting the perception they may be plesiomorphic in their anatomy—but also as sister group to a (Solifugae + Acariformes) clade at low k IW analyses, or at higher k values as sister group to all arachnids minus the ((Pseudoscorpiones + Scorpiones) + (Phalangiotarbida + Opiliones)) grouping. They appear as the earliest branching members of the equivalent clade minus fossils in extant taxa only analyses. The discovery of fossils with a plesiomorphic morphology for the group (palpigrades’ fossil record is essentially non-existent: the only fossil cannot be identified to family level) could provide key evidence to help move beyond this impasse.

Poecilophysidea

As noted above, Solifugae were traditionally allied with Pseudoscorpiones in most of the recent phylogenetic studies. Authors such as Alberti & Peretti (2002) challenged this proposal and demonstrated similarities, particularly in male genital characters, between Solifugae and the acariform branch of the mites. Two independent studies (Dabert et al., 2010; Pepato, da Rocha & Dunlop, 2010) formally recovered (Solifugae + Acariformes) based on molecular and molecular/morphological data respectively. A similar result was achieved molecular in a consensus supertree by Rota-Stabelli, Daley & Pisani (2013). This clade was formally named Poecilophysidea by Pepato, da Rocha & Dunlop (2010) drawing on a historical name used for a solifuge-like mite. Our dataset—which we concede is largely derived from the Pepato, da Rocha & Dunlop characters—also recovers Poecilophysidea under EW and three IW analyses, albeit with low support. The only exception is our IW, k = 3.0 tree where we find solifuges as the earliest branching members of a clade also containing mites and the Ricinulei. In extant taxa-only analyses there is little resolution in this part of the tree. Where recovered, Poecilophysidea synapomorphies include a testis with a distinctly larger glandular area, and presence of a nuclear envelope. In analyses including fossils, with EW and IW k = 10.0, we recover (Ricinulei + (Parasitiformes + Poecilophysidea)). In IW analyses at low k values (k = 0.25, k = 1.0), we recover the alternative (Palpigradi + Poecilophysidea); a result from Pepato, da Rocha & Dunlop (2010) upon whose study this matrix is based, and who named this clade Cephalosomata. The recovered clade (Ricinulei + (Parasitiformes + Poecilophysidea)) is a novel result but one, we note, with low support.

Ricinulei, Parasitiformes

A recent topic of debate has been the position of Ricinulei (Giribet et al., 2002; Shultz, 2007; Dunlop, Kamenz & Talarico, 2009). Traditionally, these rare arachnids were allied to the mites (Acari) on the synapomorphy of a hexapodal larva (Weygoldt & Paulus, 1979). The proposal that mites are not monophyletic has challenged this relationship and in some studies Ricinulei were placed as the sister group of the Parasitiformes branch of the mites only (e.g., Shultz, 2007). Alternatively, putative synapomorphies have been identified between Ricinulei and the extinct order Trigonotarbida, such as a locking mechanism between the prosoma and opisthosoma, tergites divided into median and lateral plates and a small terminal claw on the tip of the pedipalp (Dunlop, 1996b; Dunlop, Kamenz & Talarico, 2009). To this we could add the observation of Talarico, Lipke & Alberti (2011, Fig. 2) that ricinuleids have small filtering projections immediately in front of the mouth remarkably similar to the condition seen in well preserved trigonotarbid arachnids (Dunlop, 1994b, Fig. 4; Garwood & Dunlop, 2010, Fig. 6). One of the aims of the present study was to test whether the inclusion of trigonotarbids affects the position of Ricinulei: namely whether they are closest to one or both of the mite lineages or whether the addition of trigonotarbids modifies their position, bringing them closer to the tetrapulmonate arachnids. The result was not clear cut. With fossils in the EW analyses, and IW k = 10 we recover a (Ricinulei + (Parasitiformes + (Acariformes + Solifugae))) clade. In IW, however, at low k-values we find the Ricinulei are recovered as sister group to trigonotarbids, within the Pantetrapulmonata, on the basis of the shared characters already mentioned. In this case parasitiformes create a grade at the base of the pantetrapulmonate/ricinuleid clade, with low support values. For IW k = 3.0 we recover a (Solifugae + (Acariformes + (Ricinulei + Parasitiformes))) clade. The groupings we recover comprising Ricinulei, Parasitiformes, Acariformes and Solifugae in various arrangements are poorly-supported, and are defined by the presence of divided femora in legs 3 and 4 (albeit lost in some mite taxa), and a hinged patellotibial articulation.

In a broader sense, our analysis contributes towards the growing support that mites (Acari) are not a monophyletic group. In fact this hypothesis can be traced back to early observations by Grandjean (1935) and Grandjean (1936) that there are numerous fundamental differences in body plan between the acariform and parasitiform mites. Diphyletic origins were formally proposed by Zachvatkin (1952) and were particularly championed by Van der Hammen (1989), and references therein. Van der Hammen’s impact was limited by his rejection of cladistics and most of the early cladistic analyses treated Acari as an a priori monophyletic group (Weygoldt & Paulus, 1979; Shultz, 1990); sometimes simply scoring a generalised ‘mite’ as a terminal taxon. Subsequent studies tried to test mite monophyly by adding in a range of acariforms and parasitiforms as terminals and, as in our scheme, often recovered the two major lineages in divergent positions on the final tree (Shultz, 2007; Pepato, da Rocha & Dunlop, 2010). All mites share the putative synapomorphy of a gnathosoma—a movable unit including the chelicerae, pedipalps and mouth lips, but this single character is being increasingly outweighed by other data. Acariform mites often resolve closest to camel spiders (Solifugae; as in some trees in this study) while parasitiform mites may resolve close to ricinuleids (this study; see below) or Pseudoscorpiones (Regier et al., 2010).

Pantetrapulmonata

Shear & Selden (1986), Shear et al. (1987) and Selden, Shear & Bonamo (1991) did pioneering work on integrating the extinct order Trigonotarbida into cladistic analyses of living arachnids. Through identifying characters like the presence of two pairs of book lungs and ‘clasp-knife’ chelicerae where the fang articulates against a basal segment they concluded that Trigonotarbida is the sister group of Tetrapulmonata (see below). Further characters and character states in our analysis derived from recently generated tomographic datasets (Garwood, Dunlop & Sutton, 2009; Garwood & Dunlop, 2011; Dunlop & Garwood, 2014) continue to support this hypothesis both in the EW and IW trees, with the exception of low k-values where trigonotarbids are sister group to ricinuleids, and this clade is sister group to other tetrapulmonates. There is moderate support for this arrangement, which is otherwise not recovered elsewhere. Shultz (2007) formally named the (Trigonotarbida + Tetrapulmonata) group Pantetrapulmonata; a name we also adopt here.

Tetrapulmonata

Our data also strongly supports Tetrapulmonata (i.e., Haptopoda, Amblypygi, Thelyphonida, Schizomida, Araneae and Uraraneida); an unsurprising result as this is probably one of the least controversial clades within the arachnids (reviewed by Dunlop, Borner & Burmester, 2014). An evolutionary lineage including spiders, whip spiders and whip scorpions can be traced back in some form to early studies such as Pocock (1893) and Börner (1904), and is widely recovered from both morphological (Shultz, 2007) and molecular (Regier et al., 2010; Sharma & Giribet, 2014) datasets.

Schizotarsata

Our novel tomographic data for Haptopoda is important in that it confirms that these animals had ‘clasp-knife’ chelicerae comprising only two articles (Fig. 1B): a basal paturon and a fang. This in turn reinforces the supposition that Haptopoda belong to Tetrapulmonata, and are not Opiliones (contra Shear & Kukalová-Peck, 1990, p. 1812) in which case we would have expected chelate chelicerae with three articles. As per Shultz (2007), Haptopoda resolves in our trees as the sister group of Pedipalpi (see below). This supports his proposed clade Schizotarsata, which as before can be defined on the synapomorphy of walking legs II–IV sharing a specific pattern of three distal tarsomeres.

Pedipalpi

Amblypygi, Thelyphonida and Schizomida were originally combined as a single arachnid order, Pedipalpi (e.g., Börner, 1904). This name is now usually used to recognise a clade of these three taxa. Thelyphonida and Schizomida are unquestionably sister taxa and until 1945 were treated as a single order (e.g., Pocock, 1911). There remains debate about the position of the whip spiders (Amblypygi). Pedipalpi has a number of putative synapomorphies including an elongate first pair of legs and the raptorial pedipalps which give the clade its name. To this Shultz (1999) added numerous skeleto-muscular characters; albeit not tested across all Arachnida. The alternative hypothesis is Labellata (Amblypygi + Araneae) whereby whip spiders and spiders share characters such as a narrow pedicel between the prosoma and opisthosoma, a coalescence of the nerve ganglia in the prosoma, and a postcerebral sucking stomach. These two hypotheses and their apomorphies were compared by Alberti & Michalik (2004, Fig. 48). Pedipalpi was recovered by Shultz (1990), Shultz (2007), Giribet et al. (2002) and Regier et al. (2010). Labellata was supported by Weygoldt & Paulus (1979), Wheeler & Hayashi (1998) and Alberti & Michalik (2004). In the present analysis Pedipalpi resolves as the most parsimonious solution, with the extinct Haptopoda as its sister group as noted above. The divided prosomal shield of Schizomida is thus assumed here to be homoplastic with respect to the Cephalosomata group (see above).

Serikodiastida nom. nov.

This paper builds upon the work of Legg, Sutton & Edgecombe (2013) in formally testing the position of the extinct order Uraraneida. The Devonian genus Attercopus fimbriunguis was initially interpreted as a trigonotarbid (Shear et al., 1987) and was later proposed as the oldest spider (Selden, Shear & Bonamo, 1991). Subsequently, new Devonian material came to light which—combined with data from a Permian fossil originally thought to be a mesothele spider (Eskov & Selden, 2005)—revealed that there was a lineage of Palaeozoic arachnids which resembled spiders, but lacked spinnerets and retained a flagelliform telson rather like a whip scorpion. Selden, Shear & Sutton (2008) redescribed these fossils and proposed a new order, Uraraneida, to accommodate them. They also suggested that uraraneids were probably close to the origins of spiders, sharing with them the presence of silk spigots, but not fully developed spinnerets. Predictably, we recovered (Uraraneida + Araneae) with strong support. Araneae are united in this analysis by the presence of opisthosomal spinnerets, and (Urarineida + Araneae) by silk glands, a naked cheliceral fang and the putative presence of a cheliceral venom gland if there really is a gland pore on the fang of A. fimbriunguis. We propose here the clade name Serikodiastida to formally recognise this relationship. The name derives from the Greek serikodiastes σηρικoδιαστη´ς meaning silk worker, reflecting the shared ability of these taxa to produce silk.

Impact of fossils

One aim of this study was to include fossils with excellent and (where possible) three-dimensional preservation in a phylogenetic analysis, in order to place extinct arachnid orders. It also allows us to explore the impact their inclusion has on our understanding of chelicerate relationships. A quantitative assessment of this impact is beyond the scope of the current paper, and will be an avenue of exploration for future work. Nevertheless, we believe adding fossils to this matrix has been illustrative. It demonstrates that the addition of fossils breaks up long branches—the most extreme example being the Pedipalpi, which is defined by 28 synapomorphies in the analysis of extant taxa only, but has only three synapomorphies following the introduction of Haptopoa and Trigonotarbida. The addition of further crown group members of the constituent clades would not have this effect (see also Edgecombe, 2010; Legg, Sutton & Edgecombe, 2013). Despite the addition of fossils, long branches remain problematic in some parts of the trees presented herein—most prominently in the pycnogonids. This is an ancient group with a sparse fossil record (Dunlop & Arango, 2005; Dunlop, 2010)—accordingly the discovery of novel and complete fossil species will be key to the group’s correct placement. Recently discovered species (Kühl, Poschmann & Rust, 2013; Rudkin et al., 2013) have proven too incomplete in their preservation for confident placement of species, compounded by poor understanding of their ontogeny. The addition of fossils also results in some changes in tree topology. Stomothecata is present in the IW extant-only analysis—its absence from EW trees suggests the grouping is not stable to the addition of novel characters. With the addition of fossils in IW analyses, it is only recovered at low k values. The lack of support for Stomothecata comes about in part because two fossil scorpions and the tetropthalmid harvestman Hastrocularis argus (Garwood et al., 2014) lack a stomotheca composed of palpal and first leg coxapophyses. Similarly, the addition of fossils—most notably Offacolus kingi (Sutton et al., 2002) and Dibasterium durgae (Briggs et al., 2012)—results in a paraphyletic Synziphosura as previously discussed (Lamsdell, 2013). Such cases suggest the plesiomorphic elements present in fossils’ anatomy appear key to their placement, and this can have significant impact on our understanding of chelicerate evolution. In this example, the addition of fossils (based on IW analyses) would imply a last common chelicerate ancestor that was synziphosurine in form and so directly impacts on our model of chelicerate origins. Finally, the addition of fossils to the current analysis makes the resulting topology better resolved as shown in Fig. 3, and adds stability. A strict consensus tree of analyses conducted at k = 0.25, 1.0, 3.0 and 10.0 using just extant taxa resolves all arachnid orders in a polytomy, demonstrating the changeability of tree topology with differing concavity constants. The same tree with fossils (i.e., a consensus of analyses with varying concavity constants) in is still largely a polytomy, but there is slightly better resolution—a nested clade appears included the arachnids including all taxa non-scorpion, opilionid and pseudoscorpion taxa.

We believe this study demonstrates fossils’ utility in phylogenetic analyses—as reflected in other works (see overview by Edgecombe, 2010)—even in a group often assumed to have a patchy fossil record. However, we reiterate the caveat that even with fossils a number of our clades are poorly supported, and indeed there are still significant changes in tree topology between differing analytical parameters.

Conclusion

Adding a small number of fossils to a phylogenetic analysis of the chelicerates changes the topology of the trees recovered, reducing support for several clades, and increasing support for others. Recent decades have seen morphologically-driven cladistic analyses achieve elements of consensus—arguably in contrast to the more variable findings of molecular studies. The skeletomuscular characters of morphological analyses are, in this analysis, not robust to the addition of novel characters and fossil taxa. This instability could result from a paucity of fossils that sample the timing of cladogenesis (Legg, Sutton & Edgecombe, 2013), or from the fact that chelicerate origins lie in an ancient rapid radiation, as reported for the insects (Whitfield & Kjer, 2008). Whatever the cause, we suggest that chelicerate phylogeny—as molecular studies suggest—remains in a state of flux. Furthermore, we believe the discovery of novel fossils sampling periods closer to both chelicerate and arachnid origins will be integral to changing this.

Supplemental Information

File S1 Three dimensional reconstruction and the phylogeny of extinct chelicerate orders: Morphological characters statements.

Morphological characters statements for the 192 characters used in the current analysis, modified after Pepato, da Rocha & Dunlop (2010).

Click here for additional data file.

File S2 TNT file of current matrix

The matrix for this phylogeny, including all fossil terminals.

Click here for additional data file.

File S3 TNT file of current matrix, fossils removed

The matrix for this phylogeny, with the fossils removed.

Click here for additional data file.

File S4 Digital visualisations of the haptopod Plesiosiro madeleyi (NHM I7923), and phalangiotarbid Goniotarbus angulatus (NHM In22838)

A video showing digital visualisations of the haptopod Plesiosiro madeleyi (NHM I7923), and phalangiotarbid Goniotarbus angulatus (NHM In22838) derived from the X-ray microtomography reported herein.

Click here for additional data file.

We thank Claire Mellish (NHM), and Jon Clatworthy (Lapworth Museum Birmingham) for access to fossils in their care; David Legg for a helpful review and cladistics advice; Jo Wolfe and an anonymous reviewer for their valuable comments; and Sandra Thomas for advice on Latin and Greek vocabulary and grammar. We would like to acknowledge the assistance provided by the Manchester X-ray Imaging Facility. RG is an 1851 Royal Commission Research Fellow, a Scientific Associate at the Natural History Museum, London, and member of the Interdisciplinary Centre for Ancient Life (UMRI).

Glossary

Chelicerata s.l. pycnogonids + euchelicerates.

Euchelicerata xiphosurids, eurypterids + arachnids.

Merostomata xiphosurids + eurypterids.

Metastomata eurypterids + arachnids.

Stomothecata scorpions + opilionids.

Haplocnemata/Apatella pseudoscorpions + solifuges (widely supported).

Acaromorpha ricinuleids, acariformes + parasitiforms (widely supported).

Pantetrapulmonata trigonotarbids + tetrapulmonates.

Arachnopulmonata scorpions + tetrapulmonates.

Tetrapulmonata spiders, amblipygids, whip scorpions + schizomids. This almost certainly includes uraraneids since their publication in 2008 (Selden, Shear & Sutton, 2008).

Schizotarsata haptopods + pedipalpids.

Pedipalpi amblypygids + thelyphonids + schizomids.

Labellata spiders + amblypygids.

Cryptoperculata opilionids, ricinuleids + mites.

Dromopoda opilionids, scorpions + haplocnemataids

Poecilophysidea solifuges + acariform mites.

Cephalosomata palpigrades, solifuges + acariform mites.

Serikodiastida A name coined herein for uraraneids + Araneae.

Additional Information and Declarations

Competing Interests

Author Contributions

Data Deposition

RG is an 1851 Royal Commission Research Fellow, a Scientific Associate at the Natural History Museum, London, and member of the Interdisciplinary Centre for Ancient Life (UMRI).

Russell J. Garwood conceived and designed the experiments, performed the experiments, analyzed the data, contributed reagents/materials/analysis tools, wrote the paper, prepared figures and/or tables, reviewed drafts of the paper.

Jason Dunlop conceived and designed the experiments, performed the experiments, wrote the paper, reviewed drafts of the paper.

The following information was supplied regarding the deposition of related data:

Figshare: figshare.com/s/19e38386638c11e488d206ec4b8d1f61. The data matrix is also available in the public database Morphobank (Project 1274).

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
