# Peer review of "Three-dimensional reconstruction and the phylogeny of extinct chelicerate orders"

_PeerJ, doi:10.7717/peerj.641_

## Round 0.1 · original submission · Minor Revisions

Congratulations with this great manuscript with novel data on two additional extinct arthropod orders. The manuscript is in a good state as evidenced by minor revisions given by all 3 reviewers (a first since my start as editor here at PeerJ). However, some (minor) points still need to addressed before the manuscript can be published:

Coding of characteristics: Reviewer 1 pointed out some problems with defining and coding certain characters (including the use of habitat as a character and coding character states from different taxa); please address and integrate all the point that he raised

Implied weighting: All reviewers have raised points related with the technical (particularly reviewer 1 and 2) and philosophical issues (particularly reviewer 1) related with the use of implied weights; please address and integrate these points into your manuscripts

Data storage: Although no required by PeerJ, I think it really worth considered to uploaded your data on Morphobank or Dryad


Figure 3: I agree with reviewer 2 that your article might benefit from splitting up this figure as it is difficult to read at the moment

Uraneids: As pointed out by Reviewer 1 and Reviewer 3, Uraneids had already been included by Legg et al. (2013), so it should be at least credited as such. Your study is therefore not the first test of their position, but it lends support for this phylogenetic position.

Pycnogonids: Reviewer 2 (Joanna Wolfe) consider that the grouping and position of the pycnogonids might be partially an artefact (more details in her review). Furthermore, in the text you discuss the low number of pycnogonid fossils as a large issue. Novel pycnogonid discoveries have been made in the last years with interesting character states for some characters since (Kühl et al. 2013; Rudkin et al. 2013). It think it is rather the low number of completely preserved sea spider suitable for such analysis (see discussion in Kühl et al. 2013), so that it is rather the new discovery of well-preserved pycnogonids could have great potential to break up this branch. Please integrate these references (Kühl et al. 2013; Rudkin et al. 2013) into your study and briefly discuss these issues. It seem therefore to rephrase when more completely preserved sea spider are found.

References cited: On several occasions statements are made (some studies, the status is debated, etc.) without citing any relevant studies. At least some studies need to be cited, so the reader can verify these statements (see my comments for relevant cases).


In addition to all the points raised by the individual reviewers, please also address the following points I still noticed:

p. 2, line 4: “Chelicerates can be found back in the Cambrian”: please provide at least one reference for this statement

p. 2, line 14: “probably aquatic” should maybe be replaced with probably largely aquatic as some sea scorpions might have able to leave the water at least temporary

p. 2, line 17: “comprehensive morphological analyses”: large datasets have also been compiled for arthropods in general (e.g., Legg et al. 2013) and could also be worked into the introduction

p. 2, line 36: “Legg, Sutton & Edgecombe 2013, Sharma & Giribet 2014”: unsure if it is also necessary to cite Edgecombe 2010 in this context; it might also be valuable to cite in the context of combining molecular and fossil data

p. 3, line 11: “full range of fossil morphology”: it might be more honest to say “the fullest possible range of fossil morphology” as I am sure you had to make certain compromises

p. 4, line 16: “specimens of a related specimens”: when possible, please cite a reference who demonstrates or posits that these species are closely related (when possible)

p. 4, line 33: “extra-oral digestion, … easier to code for fossils”: it might be good to explain in more detail why this is more easy or how exactly this can be coded (as not all readers might be familiar with (fossil) arthropods)

p. 5, line 1: “reflect the state observed in outgroups and some fossil horseshoe crabs”: please give a reference for this statement if possible

p. 5, line 15: “Silurian species Haliestes… and the Devonian Palaeoisopus …”: Since the publication of Pepato et al. (2010), novel discoveries of sea spiders species have been described from the Devonian Hunsrück Slate and the Ordovician which are worth a brief remark and should be cited in this context. It for example be cited that these show interesting character states, but are to incompletely preserved to be useful for your analysis (see discussion in Kühl et al. 2013).

p. 5, line 26: “terrestrialisation events”: it is necessary to cite a reference (Garwood & Edgecombe 2011, Dunlop et al. 2013?) for readers who might not be familiar with terrestrialisation events

p. 5, line 29: “Chasmatapidids have been recovered as paraphyletic with respect to eurypterids in some studies”: you need cite at least one or better even multiple reference who had such a result as this is currently impossible to verify by the reader

p. 7, line 29: “ortognath bite”: you need to define (possible citing a reference) what an ortognath bite is as not all readers (non-arthropod workers) might be familiar with this term

p. 8, line 2: “ we treat the observation with caution”: how did you deal with this in your analysis ?

p. 8, line xx: “Some previous studies have noted possible opening for (?tracheal) spiracles among the anteriormost sternites,”: you need to cite at some of these studies as this is currently impossible to verify by the reader


p. 9, line 24: “those taxa which were traditionally assigned to Pycnogonida, Merostomata and Arachnida”: please provide a “traditional” reference who defines chelicerata in this way

p. 9, line 32: “both outgroups (A. cambricus, L. superlata) have been considered stem-chelicerates”: you need to provide references for at least some studies who considered that as such

p. 10, line 3: please replace “Xiphosurida” with “modern Xiphosurida”

p. 11, line 14: please replace “toplogy” by “topology”

p. 12, line 9: you suspect the effect of long branch attraction here (but see also reviewer 2 who suggested another explanation)

p. 12, line 10-11: “the limited number of pycnogonid taxa precludes breaking up this long branch with such fossils”: this might be the case, but it is more complicated than that: new pycnogonids are discovered (some with rather interesting character states: Kühl et al. 2013; Rudkin et al. 2012) almost each year, but there are often too incompletely preserved to be relevant for phylogenetic analysis: it would therefore speak of “the limited number of completely preserved pycnogonid taxa” instead

p. 12, line 27: you cited Tetlie (2007) in this context, but as far as I know he only used a handmade tree,; this needs to be at least clearly stated (see also comments by reviewer 1)

p. 13, line 4: “Opiliones a recovered” should be replace with “Opiliones are recovered”

p. 13, line 14: should it be “Solifugae” instead of “Solifuge”

p. 14, line 28 : “A recent topic of debate has been the position of the Ricinulei”: you need to cite at least some articles which have debated their position

p. 14, line 30: “Traditionally, they were allied to mites (Acari)”: please give some references who allied them to the mites based on their larva and gnathosoma

p. 15, line XX: you could replace “growing hypothesis” also by “growing support”

p. 15, line 22: “early cladistics analyses treated Acari as an apriori monophyletic group”: you need to provide at least some references of these early studies

p. 15, line 24: “Subsequent studies … often recovered two major lineages in divergent positions on the final tree”: you need to provide at least some references of these subsequent studies

p. 16, line 17 (pedipalpi): “until 1945… treated as a single order”: you need to provide a reference for this study or work which treated them as a single order

p. 16, line 29 (Serikodiastida): “we offer the first formal test of the position of the extinct order Uraraneida”: this is not entirely true as Attercopus was also included in the analysis of Leg et al. 2013 (see comments by reviewer 1 and 3): this needs to be rephrased

p. 17, line 4: “accordingly the discovery of novel species will be key to the group´s correct placement”: you should also mention the “novel species based on complete material” as several new species have recently been described (e.g., Kühl et al. 2013, Rudkin et al. 2013), but are probably too incompletely preserved to be relevant: please cite these works and discuss this problem in an additional sentence as it is an important statement to make

p. 17, line 11: “paraphyletic Synziphosura”: please cite Lamsdell 2013 once more in this context for completeness sake

p. 17, line 19: “Edgecombe 2010” is not yet listed in the references, please verify if all references are cited in the reference list

p. 58, character 190. Habitat: it might be good to explain on what basis a terrestrial mode of life has been or can be inferred for Paleozoic scorpions (cf. discussion in Kühl et al. 2012 and maybe some additional reference who debated it). Please state on what the terrestrial mode of life of Paleoscorpius and Compsoscorpius is based on (presence of book lungs?, facies distribution, etc.)

Reviewer 1 ·

Basic reporting

The manuscript presents a new analysis of two extinct arachnid orders utilizing data from fossil specimens imaged with X-ray microtomography combined with a broad morphological phylogenetic analysis of all the chelicerate orders. The results are of great interest to all arthropod workers, and have further ramifications for a number of fields relating to molecular phylogenetics and divergence times, the number of terrestrialization events that have occurred and the timing of these events, and the evolution of parasitism. This therefore represents an important and worthwhile contribution assessing our current state of knowledge of chelicerate relationships.

The manuscript is appropriately formatted and well presented, providing an extensive background that serves to put the current work in context and provides comprehensive citation of previous studies. The figures are clear, well labeled, and in an appropriate resolution. There are some minor typographical errors scattered throughout the paper that can be easily corrected, but nothing beyond the usual for a paper of this size and scope.

Experimental design

The authors have a wealth of experience in generating and studying 3D models from X-ray microtomography of fossil specimens, and the specimens presented here appear to have been expertly scanned. The authors are very explicit about the limitations of the specimens, and refrain from over-interpretation of the available material.

There are a number of issues surrounding the cladistic analysis which the authors should be made aware of, both surrounding the formulation and treatment of characters and the subsequent presentation of the results (which is discussed in the next section). There are two main issues treated here; the first revolves around the manner in which some characters have been formulated and coded, either in general or for specific taxa, while the second centers around the use of implied weights and the technical and philosophical issues that come with it.

In regard to character coding, there are a number of points that the authors need to address:
- The authors code Alalcomenaeus cambricus as the outgroup, however they also utilize data from the Tanaka et al. paper describing neural structures in Alalcomenaeus sp., which is likely not the same species as the Burgess Shale taxon.

- The authors state they coded Parastylonurus hendersoni for the analysis, however it appears that limbs are not known from this species and it seems likely that the authors have in fact coded Parastylonurus ornatus instead.

- Offacolus is coded as not having ophthalmic ridges, however Sutton et al. (2002; Proceedings B) described a number of raised and depressed structures on the carapace, one of which is in the correct position for the ophthalmic ridge. While coding an ophthalmic ridge in Offacolus may at present be a bit of a stretch, perhaps having it as uncertain may be appropriate.

- The authors state that there is no apomorphy for defining eurypterids, however Lamsdell (2011; Palaeontographical Society monograph) suggested that the fusion of the first two abdominal appendages into the genital operculum may be a convincing autapomorphy. This may be a character worth including in the phylogenetic analysis.

- Leanchoilia is coded as character state 1 for character 82 (appendages on opisthosomal segment 1: lost). No such indication is given in the detailed work of Haug et al. (2012; BMC Evolutionary Biology) and so it seems probable that this coding was erroneous.

- Is character 90 (habitat) truly a valid character to include in the analysis? Various terrestrial or aquatic adaptations are already coded based on morphology, and so this seems likely to be simply biasing the analysis by up-weighting terrestrialization characteristics. Furthermore, it is unclear how valid assuming homology for ecological changes really is, especially given current research into niche conservatism (Stigall 2014; Ecography), and we know that terrestrialization events have occurred numerous times across different groups.

- Character 18 (ingestion of solid material) is again likely extraneous; it is correlated with character 52 (presence or absence of gnathobases) and appears to just be up-weighting preoral digestion.

- Character 70 (plate-like opisthosomal appendages); the authors cite Jeram (2001) in regard to scorpions, but should probably also cite Farley (2001; Invertebrate Reproduction and Development).

- Character 93 (presence or absence of eyes) can be merged with character 94 (presence or absence of lateral eyes). The authors state that they have created this character to avoid up-weighting total loss of eyes, however they still code character 97 (presence or absence of median eyes) as absent when the species is totally blind. It is probably best to assume independence of these two characters.

- Character 9 (cephalic doublure) is coded as absent for Dibasterium and Offacolus, however given that the specimens are external molds within ash nodules and the doublure is preserved flat on the ventral body wall it is unlikely that the doublure would be visible if it were present. It is probably best to code this character as unknown in these taxa.

- The coding of character 32 (presence of elbowed chelicerae) seems a bit haphazard. Weinbergina is coded as having an elbow joint, although none of the described specimens convincingly show the chelicera in detail – in fact multiple interpretive drawings from Sturmer and Bergstrom (1981; Palaontologische Zeitschrift) reconstruct the chelicerae as projecting directly forward from the carapace. Both Offacolus and Dibasterium are coded as not possessing an elbow joint, yet both taxa clearly have chelicera that project posteriorly with a marked angle at one of their podomere joints; this seems to fit the diagnosis of an elbow joint. None of the pycnogonids are coded as having elbow joints, while there is a distinct angle between the peduncle and the subsequent cheliceral podomeres; the authors may have decided this does not represent a true elbow joint, but if so this should be made clear with justification. Perhaps most perplexing is that both Alalcomenaeus and Leanchoilia are coded as lacking an elbow joint, despite multiple papers demonstrating its presence in megacheirans (Chen et al. 2004; Haug et al 2012 a,b).

Implied weights have not been widely adopted among the biological sciences, and it is somewhat puzzling that they have begun to find a following among paleontologists; perhaps problems with missing data, which is less of an issue in biology. From a purely philosophical perspective, it should be made clear that any form of character weighting, including equal weighting, makes assumptions about the evolutionary process. Differential weighting, however, appears to assume more than equal weighting as it implies that the investigator knows the relative behavior of one kind of transformation relative to another; as such it is a form of model selection, however there is no statistical basis for favoring one type of model over another. Implied weights in particular have been criticized for being unpredictable and depending strongly on the concavity factor k, for which there is no biological reason for preferring one value over another; of equal concern is the fact that implied weights frequently retrieve topologies that are not actually found in any of the most parsimonious trees under equal weighting (Turner and Zandee 1995; Cladistics). This violates the basic framework of parsimony and suggests that the model adopted by implied weights is not strictly parsimonious; the parsimony model is used predominately in morphological analyses as it is impossible to know the frequency with which character reversals occur. Goloboff’s (1993, 1995; Cladistics) standpoint that parsimony applies given the appropriate weight of the characters is therefore hard to justify, as there is no easily defensible reason for differential weighting of characters; which characters are ‘more important’ in evolution is an unknowable, as is the true amount of homoplasy exhibited by a morphological characteristic, i.e. we cannot tell what the appropriate weight of a character is, and it is therefore more conservative to weight them all equally.

Implied weighting also makes a number of assumptions about character sampling and coding; under implied weighting, accidental miscoding of a character can have important ramifications for character weighting. Implied weighting also fundamentally conflicts with Hennig’s Auxiliary Principle; to assume homology if there is no evidence to suggest otherwise. The result of this is that in assuming homology of such characters an investigator is likely to introduce homoplasy into an analysis (with the overall total weight of evidence from all the included characters informing whether the statement of homology is justified) that will result in implied weights down-weighting the character. This can be problematic as the character may itself define clades even though it is homoplasious; down-weighting the character will therefore result in lower support for these clades. It is worth noting that homoplasy has been shown to increase phylogenetic structure among nucleotide sequences in exactly the same manner as just described (Kallersjo et al 1999; Cladistics) – a fact that has been ignored by proponents of implied weights, although Goloboff (in press; Cladistics) has recently enacted the ability to apply implied weights to partitions of the dataset because of this very issue – and there is no reason why the same should not apply to morphological datasets. Investigator-added homoplasy is also going to influence the overall F values of a topology, which will in turn influence how implied weighting is enacted. Missing and inapplicable characters are also a problem; characters with a large number of missing or inapplicable codings are likely to exhibit less homoplasy simply because they are coded for less taxa. This again influences the F value and can result in characters with a high number of inapplicable or missing codings being up-weighted in the analysis.

It is also worth noting that homoplasy can only be recognized after a tree topology is obtained. While implied weighting utilizes methods concurrent with tree searching in order to avoid the recursive nature on basing weighting on a topology which is based on weighting, it is worth remembering that an initial topology must be generated with which to infer homoplasy, and there is therefore potential for suboptimal plateaus in treespace to influence the results. There is also the danger that implied weights may up-weight homoplasious characters at the expense of homologous ones for trees with very high degrees of homoplasy; this can result in spurious topologies that nonetheless appear well supported. Relying on implied weights for retrieving tree topologies is therefore not as reliable or straight forward as has been assumed, particularly as adding or removing constant or autapomorphic characters has been shown to influence the topologies retrieved under implied weights, presumably through perturbations of the F value.

The authors apply implied weights as a form of stress testing to see how homoplasy influences the topology. This is perhaps the most conservative way to apply implied weights, and may provide useful information about the mutability of the tree topology given different weighting parameters. The way this is represented in the manuscript however could be improved; the authors present a majority rule consensus whereas a strict consensus would be better (see more on this below) and do not incorporate the equal weights most parsimonious trees into the consensus. Showing a strict consensus of all most parsimonious trees derived from the equal weights and implied weights analysis of different concavity factors will show which nodes are most robust to perturbations in model parameters, however implied weighting under a particular concavity factor does not in itself provide stronger support for a topology than equal weighting. Given that implied weighting violates a number of the basic principles of parsimony-based phylogenetics, it is best to treat results derived from this method with extreme caution and to present them in as conservative a way as possible, which in this case would be through a strict consensus als

Validity of the findings

The presentation and discussion of the findings is generally valid, however there are a few points that should be addressed. The authors briefly mention that megacheirans are resolved as a grade in their analysis, but appear to have failed to realize that this could be because TNT only allows for a single outgroup. There is therefore no way that the analysis would resolve megacheirans as a clade and will force them to be paraphyletic. The current analysis therefore provides no information regarding the status of megacheiran monophyly.

Finally, the use of majority rule consensus trees is ill advised. Majority rule consensus trees are generally only reported for parsimony analyses when the strict consensus is poorly resolved. Majority rule consensus trees are however meaningless under the parsimony optimality criterion as there still exist equally parsimonious trees that are equally valid alternatives to the majority rule consensus. The retrieved most parsimonious trees do not represent a randomly sampled normal distribution and therefore the percentages retrieved are meaningless; there is no reason to suppose that the percentage of times a particular tree resolution appears in a result has any significance as quite different results can be equally parsimonious. Furthermore, the strict consensus for the equally weighted analysis is rather well resolved (see attached file), and the authors should simply report this result. The majority rule tree differs only in increasing resolution of the Acariformes, and as this is not the focus of the paper its inclusion is not really necessary. The authors also fail to put the percentage values for each node on the majority rule tree.

The use of the majority rule tree to combine analyses of different k values for the implied weights analysis is also unusual. While the concept (to show the points of agreements between the different analyses) is understandable, as stated earlier, a strict consensus would again be a more conservative and transparent way to do this.

Additional comments

The manuscript is a well put-together and formulated study with important results. The points raised here should not negatively impact the results of the study in any way; presenting strict consensus trees are not going to result in a massive loss of resolution, and the changes in the character matrix are most likely minor (although they should be evaluated an enacted where necessary). The authors should also add some discussion regarding the potential drawbacks and problems of implied weights, which is something that no study that uses the method has yet done.

The only other two minor things that should probably be addressed is that the analysis of Tetlie (2007) that the authors use to argue about the placement of various eurypterids is not actually the result of a phylogenetic analysis but simply a handmade tree that takes information from a number of published phylogenetic analyses – none of which include the group which the authors are particularly interested in the placement of. It may be more appropriate for them to find another analysis that has included these taxa, or state that the Tetlie paper is not actually the result of a phylogenetic analysis.

Finally, the authors also state that they present the first formal test of the uraraneids, however Legg et al (2013; Nature Communications) also included Attercopus in their analysis, resolving it in the same position as the current study.

Overall an excellent contribution, which with consideration of the points raised here will be most suitable for publication.

Annotated reviews are not available for download in order to protect the identity of reviewers who chose to remain anonymous.

·

Basic reporting

The ms is appropriate for publication based on these criteria. It is a broad study including morphological description of two fossils and a cladistic analysis testing their position in the chelicerates.

However, Figure 3 (in the pdf) is too low-res to read. It is ok in the overview file. Perhaps it would be good to split the equal and implied weights figures regardless to make them bigger/more readable.

Experimental design

The methods are well defined and technically sound. It would be nice to see (even in the supplement) a comparison to the strict consensus tree, which is more common for morphological publications.

Missing data is somewhat discussed, but the authors did not measure its effect per se. A breakdown of % missing data per species by fossil and extant should be easy to compile. In particular, I would also like to see the so-called ‘wildcard’ taxa identified (the ones that had huge impact on IW analyses) to see if it was missing data or straight homoplastic codings that influenced their varying positions (or if these taxa were fossil vs. extant).

Validity of the findings

The conclusions are appropriately stated and statistically sound, however, the authors may note the following:

I think the position of pycnogonids is sort of forced by the taxon sampling herein. First, pycnos + euchelicerates is automatically assumed, though Legg et al. (2013) showed Cormogonida was a possible result when fossils were excluded from the matrix. As this study also tests the contribution of fossils, it might be appropriate to include a few mandibulate terminals besides just megacheirans (with that said, I do recognize the difficulty in applying the existing characters). While the authors suggest LBA as the source of the pycnogonid position, it is not really tested (e.g. with additional outgroups, as mentioned). Where mtDNA phylogeny is mentioned, also please consult Simon & Hadrys (2013, MPE) for a thorough damnation of mtDNA for deep divergences in arthropods.

I wouldn’t bury the comment that the new data for Haptopoda resolve their cheliceral morphology and therefore strengthen support for their position. A stronger statement could be made, even mentioned in the abstract.

Finally, although not specifically required by PeerJ, it would significantly increase the impact of the ms (and reusability of the associated data) if the matrices, MPTs, and consensus trees were deposited in an online database, such as MorphoBank and/or Dryad.

Additional comments

Minor corrections (sorry no line numbers, the pdf doesn’t have them):

- Page 2: Rehm misspelled
- Page 3: “numerous” new fossil taxa and characters – why not just say how many?
- Page 4: “current study has different goal” should be “goals”
- Page 4: Bercovici et al. mentioned but not in references list at the end
- Page 6: Selden, Shear & Sutton 2009 mentioned but not in references list at the end
- Page 8: “resolvability” or “resolution”?
- Page 9: Very cosmetic, but I would introduce the EW and IW abbreviations in the methods section when weighting is first mentioned, rather than at the beginning of the discussion.
- Page 11: use of exclamation point too informal
- Page 11: Shultz misspelled
- Page 13: phalangiotarbid misspelled
- Page 15: Rota-Stabelli misspelled
- Page 16: Shultz misspelled
- Page 16: so this is not a synapomorphy anymore, as it fails the test of primary homology, it is at best a symplesiomorphy (depending on its distribution?) and more likely just homoplastic.
- Page 17: Shultz misspelled
References:
- Dunlop & Garwood APP needs a year (2014 or in press?)
- There are multiple references to Dunlop (1994) and Dunlop (1994b), so probably the first one needs an (a).
- A few references in the list which are not referred to in the text: Haase 1890; Karsch 1882; Petrunkevitch 1913, 1945.
- Fig 3: is Pycnogonida a basal polytomy (i.e. potentially paraphyletic) when fossils are excluded? That’s what the figure seems to show but is not clear from text. Certainly they are monophyletic in the fossils-included trees.
- Fig. 3: also, is that the majority rule consensus of all IW analyses with all the different k-values? In my experience higher k = more resolved tree (because it is essentially downweighting homoplastic characters), so it is pretty surprising to get such consistent results. Might be worth a brief comment.

·

Basic reporting

No comments

Experimental design

No comments

Validity of the findings

No comments

Additional comments

Review of Garwood and Dunlop (PeerJ)

Overall this is an exceptional and well written paper and I would definitely recommend it for publication. All issues pertaining to this manuscript (see below) are minor and should not prohibit publication of this article.

General notes:

The term ‘carapace’ is used throughout the manuscript to describe the anterior synsclerite of chelicerates. I would personally caution against the use of this term as a carapace (in crustacean terms) refers to the posterior extension of a particular segment (usually maxillary) which may later fuse to the anterior trunk segments (Olesen 2013). However the chelicerate “carapace” is a synsclerite resulting from the fusion of an anterior lobe and (at least) four posterior tergites (Liu et al. 2010). The term prosomal shield might also be used but excludes comparison with non-chelicerates. Perhaps cephalic shield might be used instead.

Specific notes:

Page 5 (within Taxon selection):
Edgecombe et al. (2011) is cited as evidence for a close relationship between chelicerates and megachierans. This paper does not present any novel data for this hypothesis however and the results of their phylogenetic analysis were discussed in another paper (Legg 2013) which concluded that a sister-taxon between megacheirans and chelicerates in this, and other phylogenenetic analyses, is caused by inappropriate outgroup selection. Perhaps (if necessary) cite Chen et al. (2004), or Dunlop (2005).

“Chasmataspidids have been recovered as paraphyletic with regards to eurypterids in some studies”. – Could add a citation or two to back up this statement.

“Legg, Sutton & Edgecombe (2012)” should be “Legg et al. (2012)” in relation to Compsoscorpius.

Page 6 (within Taxon selection).
Trigonotarbids has been misspelt on line 2.

Is there a paper which discusses the content of Palaeocharinus? If not then in might be an idea to say, even informally, “we have coded Palaeocharinus based on specimens assigned to [species x] and [species y], which we consider to be synonymous”.

Page 6 (within Cladistic analysis)
Perhaps cite Goloboff et al. (2008; Weighting against homoplasy improves phylogenetic…) and Legg and Caron (2014; New Middle Cambrian bivalved arthropods…) which give a more extensive discussion on the use of implied weighting.

I would recommend using Symmetric resampling rather than the other methods of measuring group support in this study. Jackkniffing, bootstrapping, and bremer support are all affected by character weighting schemes and therefore not reliable for trees produced using implied weighting. In such instances Symmetric resampling should be used.

Page 7 (within Reconstruction of Plesiosiro madeleyi)
The term “taphonomic artefact” covers all manner of sins – could you be more specific about what has actually happened?

Page 9 (within Cladistic Results and Discussion)
Report tree values for implied weighted trees. If you want a way of comparing lengths across different weighting schemes then you can use a weighted character fit.

Page 9 (within Chelicerata)
The two megacheirans do not form a grade. As one of them represents a prime outgroup their relationships to one another cannot be determined.

Might I suggest that Legg et al. (2012; Cambrian bivalved arthropod reveals…) and Legg (2013; Multi-segmented arthropods..) are also cited at the end of this paragraph as they both contain a discussion of character acquisition pertaining to Megacheira. [Sorry for being “that guy”!]

Page 9-10 (Merostomata?)
This is more of a general comment but I think the presence of Merostomata is an artefact of rooting, especially considering the presence of a cephalic doublure is a character supporting this relationship. Because megacheirans share so few characters with chelicerates (GAs not withstanding) there are very few characters to actually polarise relationships within Chelicerata and therefore we would expect to find analytical artefacts within basal members. This was discussed by Gauthier et al. (1988) and Legg (2013 – to an extent). At the moment characters linking scorpions and eurypterids are resolving as synapomorphies of Chelicerata and putative “primitive” features observed in xiphosurans are not being driven downwards – a way of testing this hypothesis would be to use an artiopodan (such as Emeraldella or Sidneyia, or even Aglaspis) to see how they affect relationships within Chelicerata.

Page 14 (within Palpigradi)
A flagelliform telson is listed as a plesiomorphic characters states. Within Arachnida I would argue this is actually a derived character state as it is shared by many members of Tetrapulmata.

Page 17 (within Serikodiastida)
IT’S ABOUT TIME THIS CLADE HAS A NAME!!! Remember to capitalise it on page 17 though.

Page 17 (within Glossary)
Tetrapulmonata – “no cladistics analysis before this has included [uraraneids]” – besides Legg et al. 2013 (see fig. 4g) of course ;)

---

## Round 0.2 · Minor Revisions

Dear authors,

Thanks for integrating all the requests of the reviewers and following my additional recommendations. The paper is as good as accepted and needs no additional reviews. Your paper will be accepted pending upon the resolution of some minor formatting problems and addition of some missing references from the reference list. These minor things refrain me from accepting the current version, because no additional changes can be made to the manuscript after acceptance (through the system by me). I apologize for the inconvenience and thank you for your understanding. Please change/update:

line 12: do you mean Dunlop, Anderson & Braddy 2003 or 2004 ? Please change to 2004 as listed in the reference or add this additional reference
line 194: "opaque ontogenetic stage": please rephrase as it is not entirely clear what you mean to say here (later in the text this is phrased in a better way: line 874)
line 235: please add "Hirst, 1923" to the reference list
line 255: please add the morphobank web url within the brackets as the readers of PeerJ might not be familiar with it
lines 380-381: please add Dunlop & Horrocks 1997 to the reference list as i could not find it
line 619: do you mean that Savory was just "using" this term or did he also introduce it ?
line 700: please add "molecular" to "consensus supertree"
line 943: i would delete analysis after equal weights and add "(EW)" after it as this abbreviation is commonly used in the text
ine 949: i would add "(IW)" after "implied weights" as this abbreviation is commonly used in the text
line 2882-2883: please check this references as in the PDF I could only recognize boxes instead of the journal name (I suspect it has a Japanese name)

---

## Round 0.3 · accepted · Accept

Thanks for replying so quickly and implementing these last changes. It has been a real pleasure to handle your manuscript.